# Breathwork Interventions for Adults with Clinically Diagnosed Anxiety Disorders: A Scoping Review

**DOI:** 10.3390/brainsci13020256

**Published:** 2023-02-02

**Authors:** Blerida Banushi, Madeline Brendle, Anya Ragnhildstveit, Tara Murphy, Claire Moore, Johannes Egberts, Reid Robison

**Affiliations:** 1Faculty of Medicine, University of Queensland, Diamantina Institute, Brisbane, QLD 4102, Australia; 2Department of Pharmacotherapy, University of Utah College of Pharmacy, Salt Lake City, UT 84112, USA; 3Numinus Wellness, Draper, UT 84020, USA; 4Integrated Research Literacy Group, Draper, UT 84020, USA; 5Department of Psychiatry, University of Cambridge, Cambridge CB2 0SZ, UK; 6Department of Psychology and Neuroscience, Duke University, Durham, NC 27708, USA; 7Department of Psychiatry, University of Wisconsin Hospitals & Clinics, Madison, WI 53792, USA; 8Breathless Expeditions, Manly, NSW 2095, Australia; 9Department of Psychiatry, University of Utah School of Medicine, Salt Lake City, UT 84108, USA

**Keywords:** breathwork, pranayama, diaphragmatic breathing, hyperventilation, heart rate variability biofeedback, anxiety disorders, agoraphobia, anxiety, stress, phobia

## Abstract

Anxiety disorders are the most common group of mental disorders, but they are often underrecognized and undertreated in primary care. Dysfunctional breathing is a hallmark of anxiety disorders; however, mainstays of treatments do not tackle breathing in patients suffering anxiety. This scoping review aims to identify the nature and extent of the available research literature on the efficacy of breathwork interventions for adults with clinically diagnosed anxiety disorders using the DSM-5 classification system. Using the PRISMA extension for scoping reviews, a search of PubMed, Embase, and Scopus was conducted using terms related to anxiety disorders and breathwork interventions. Only clinical studies using breathwork (without the combination of other interventions) and performed on adult patients diagnosed with an anxiety disorder using the DSM-5 classification system were included. From 1081 articles identified across three databases, sixteen were included for the review. A range of breathwork interventions yielded significant improvements in anxiety symptoms in patients clinically diagnosed with anxiety disorders. The results around the role of hyperventilation in treatment of anxiety were contradictory in few of the examined studies. This evidence-based review supports the clinical utility of breathwork interventions and discusses effective treatment options and protocols that are feasible and accessible to patients suffering anxiety. Current gaps in knowledge for future research directions have also been identified.

## 1. Introduction

Anxiety disorders constitute the most prevalent psychiatric conditions in the world, accounting for 28.68 million years of healthy life lost due to disability or premature death [1]. In the Global Burden of Diseases, Injuries, and Risk Factors Study of 2016, the World Health Organization (WHO) ranked anxiety disorders as the ninth most common cause of health-related disability [2,3,4]. The Diagnostic and Statistical Manual of Mental Disorders, Fifth Edition (DSM-5), classifies anxiety disorders based on condition-specific symptoms, broadly including excessive fear (i.e., an emotional response to an imminent threat) and anxiety (i.e., the anticipation of a future threat), as well as related behavioral disturbances (e.g., restlessness, fatigue, and insomnia). Anxiety disorders and phobias in the DSM-5 are comprised of generalized anxiety disorder (GAD), panic disorder (PD), agoraphobia (AG), separation anxiety disorder (SAD), selective mutism (SM), specific phobia (SP), and social anxiety disorder (SAD) in addition to substance-/medication-induced anxiety disorder, anxiety disorder due to another medical condition, other specified anxiety disorder, and unspecified anxiety disorder.

While anxiety disorders differ in their symptomatology, commonly shared features include maladaptive attention bias to a threat, even when the threat is irrelevant [5]; fear generalization, in which fear responses generalize or become associated with related stimuli, such as objects, people, situations, or thoughts, subsequently resulting in excessive and unrestricted fear as well as elevated anxiety [6]; and persistent avoidance that alleviates anxiety in the short-term, yet maintains fear and anxiety in the long term, facilitating maladaptive beliefs [7,8,9]. Heightened anxiety over protracted periods can lead to a wide variety of physical symptoms and behavioral changes, such as shortness of breath, palpitations, insomnia, and restlessness. These can have severe implications for overall health and well-being [10].

Despite their prevalence and disease burden, anxiety disorders remain under-diagnosed and under-treated [11,12,13]. Furthermore, when treated, the quality of care is often poor [14], with substantial patient dissatisfaction. This results in a perceived unmet need [15]. In a representative U.S. sample, the rate of appropriate treatment, indicated by patients following treatment recommendations consistent with clinical practice guidelines, was very low [16]. Specifically, 80% of adults with anxiety disorders sought treatment through primary care physicians; however, only ~20% received appropriate pharmacotherapy, and ~10% received appropriate counselling [16]. In addition to the poor quality of care, depression and anxiety can directly and negatively impact patient satisfaction in resident clinic settings.

Integrated approaches combining pharmacotherapy and psychotherapy are the mainstays of treatment for anxiety disorders [17]. Yet, the financial burden of these interventions is staggering, with a predicted cost of USD 56 billion from 2016 to 2030 [18]. Pharmacological treatments have shown efficacy in relieving anxiety in clinical trials; however, medications such as benzodiazepines and paroxetine are poorly tolerated, leading to adherence and compliance issues [19].

Although pharmacological and psychotherapeutic options are available, many patients do not seek treatment and often face practical limitations, such as service cost, undesirable side effects, or lack of access to providers, and prefer to self-manage their condition instead [17]. Hence, there is a need for cost-effective, non-pharmacologic, and self-administered therapies that can be utilized by large populations to relieve stress and anxiety.

Respiratory abnormalities, such as hyperventilation and unexplained dyspnea, are hallmarks of anxiety and panic, and several anxiety disorders are associated with altered patterns of breathing [20,21]. The most well-known expression of this is hyperventilation syndrome (HVS) [22], characterized by various psychological and somatic symptoms induced by rapid or deep breathing. This is often linked or secondary to anxiety disorders [23,24]. During hyperventilation, the arterial partial pressure of carbon dioxide (pCO_2_) falls below 30 mmHg and rises above 7.4 pH. This reduces cerebral blood flow, which can lead to disturbed conscious awareness as well as neuromuscular irritability, electrocardio-graphic changes, tachycardia, and frequent arrhythmia [25,26]. The amygdala has been shown to play a role in the CO_2_-induced fear, and patients clinically diagnosed with anxiety have smaller amygdalae and a higher hypersensitivity toward CO_2_ [27].

Targeting dysfunctional breathing in those with anxiety is therefore expected to have a clinically significant therapeutic effect, thereby improving hyperventilation and hypocapnia, decreasing thorax and shoulder muscle tension, and increasing parasympathetic activation that strongly relates to stress response, emotion regulation, and neuro-endocrine function associated with anxiety [20,21,28,29,30,31].

Breathing practices, otherwise referred to as breathwork or pranayama, have a long and varied history in many Eastern traditions. They involve voluntarily changing the rate, pattern, and depth of respiration, affecting both cardiac and cortical activity [32,33]. Various forms of breathing practices exist, such as deep breathing, diaphragmatic or belly breathing, pursed lip breathing, and glossopharyngeal breathing. In psychiatric research and clinical practice, breathwork has been shown to diagnostically improve symptoms of anxiety, depression, trauma, addiction, obsessions, compulsions, and inattention [34,35,36,37]. Different breathing techniques have specifically been effective in reducing levels of stress [20]. Among these, controlled breathing, slow breathing, and diaphragmatic breathing have yielded the most patient benefit [38,39,40,41].

A range of underlying mechanisms have been proposed in the attempts to understand the physiological effects of slow and diaphragmatic breathing, including the polyvagal theory and the increase in parasympathetic activity through heart rate variability (HRV) [42,43,44]. In a simplistic model, the two arms of the autonomic nervous system (NS), the sympathetic and parasympathetic, have opposite effects on the heart. Stimulation by the sympathetic system nerves, which occurs during stress or the so called “fight-or-flight” response, result in an increase in heart rate. However, parasympathetic activity is most active under restful conditions (i.e., “rest and digest”) and results in a decrease in heart rate via the vagus nerve [42] (Figure 1). Breathing has a strong effect on heart rate and blood pressure, known as cardiorespiratory coupling [45], and deep slow breathing exercises have been shown to improve heart rate variability [46]. Therefore, slow diaphragmatic breathwork practices have profound effects not only on breathing efficiency but also on the cardiovascular function and autonomic functions of the NS, and these effects are bidirectional (Figure 1). The bidirectional relationship explains why hyperventilation or fast breathing are symptoms of anxiety and panic disorders. However, temporarily and voluntarily induced stress, such as short-term stress induced by exercise, fast breathing techniques, or cold exposure, can have long-term positive effects on reducing stress and improving mental health [47,48,49] and have also been shown to voluntary activate the sympathetic nervous system and suppress the innate immune response [50].

Short-term stress shall be distinguished from chronic stress that is associated with poor mental and physical health [51,52].

Despite a strong evidence base supporting the utility of breathwork for anxiety in the clinical population, no reviews—to the best of our knowledge—have systematically mapped the literature on this topic to date. As such, the aim of this scoping review was to characterize breathwork interventions for adults clinically diagnosed with an anxiety disorder, emphasizing treatment regimens and efficacy. A scoping- or exploratory-based approach was chosen given study heterogeneity. Only non-randomized and randomized studies were included to provide the highest internal validity and levels of evidence available. Likewise, all types of breathwork practices and techniques, applied as primary interventions, were included to offer a comprehensive summary of approaches used in the field. Changes in physiological, psychological, and behavioral parameters were additionally examined.

The results of this scoping review will serve to identify key trends, current gaps, and areas for further inquiry regarding breathwork interventions for anxiety disorders. Taken together, this will inform future research and clinical practice.

## 2. Materials and Methods

This scoping review adhered to the Preferred Reporting Items for Systematic Reviews and Meta-Analyses Extension for Scoping Reviews (PRISMA-ScR) guideline). The Joanna Briggs Institute (JBI) methodology and stage schema for scoping reviews were incorporated to include defining the research question; identifying relevant articles; selecting eligible studies; charting the data; and collating, summarizing, and reporting the results. The review protocol was registered a priori through the Open Science Framework (OSF), Identifier osf-registrations-a768w-v1. The 2020 PRISMA-ScR checklist is available for consultation in the Appendix A.

The PubMed (National Library of Medicine), Embase (Elsevier), and Scopus (Elsevier) electronic databases were queried on May 19, 2022, for relevant articles. Boolean operators “AND” and “OR” were used as conjunctions to combine keywords related to breathwork interventions and anxiety disorders, using DSM-5 classification and terminology (Appendix A). Study type, age range, and language restrictions were applied, refining by clinical studies, young to older adults, and English, respectively. The search terms used to identify articles are provided in Table 1, with the full search strategy and outputs per database presented in Appendix A.

Inclusion criteria consisted of (1) adults diagnosed with an anxiety or stress-related disorder, according to the DSM-5, with or without prior use of breathwork techniques; (2) breathwork techniques investigated as a primary intervention; (3) non-randomized or randomized clinical studies; and (4) quantitative and/or qualitative outcome data, including physiological (e.g., skin conductance, pCO_2_, and tidal volume [VT]), cognitive or behavioral (e.g., anxiety, apprehension, and panic attacks), and/or soft (e.g., psychosocial function and quality of life) endpoints. Exclusion criteria comprised (1) breathwork techniques without a voluntary modulation of breathing; (2) breathwork techniques combined with movement (e.g., exercise or yoga) or other non-psychotherapeutic interventions (e.g., dietary changes or supplementation, meditation, or visualization); (3) patients not clinically diagnosed with an anxiety or stress-related disorder according to the DSM-5 classification system; and (4) student theses or dissertations, conference abstracts or articles, methodology or technical note papers, editorials, commentaries, or opinion pieces, and review articles of any kind (i.e., narrative, systematic, and scoping reviews as well as meta-analyses). Duplicates were excluded prior to screening, and studies that failed to meet full inclusion criteria were excluded from the overall analysis. The Population, Intervention, Comparison, Outcomes, and Study Design (PICOS) framework was applied for evaluating eligibility in the qualitative synthesis (Table 2).

Two independent reviewers (M.B. and C.M.) screened titles and abstracts against the inclusion criteria of all articles identified in the search. Relevant articles were then selected for full-text screening and assessed for eligibility. The reference lists and citations of articles were additionally screened to identify other potentially eligible studies. A third independent reviewer (B.B.) compared and reported inter-rater agreements, reconciling any disputes. Another set of independent reviewers (A.R. and T.M.) subsequently extracted data from eligible studies into a Microsoft Excel spreadsheet (Microsoft Corp., Redmond, WA, USA). Table cells were labelled as “not applicable” (N/A) if parameters of interest were missing. The lead investigator (B.B.) performed quality assessment checks to ensure global data integrity. Data extracted from eligible studies included the study year, study design, clinical diagnosis, patient characteristics, breathwork technique, breathwork regimen (duration, frequency, period, and follow-up), outcome measures (objective and subjective), results and limitations, and any other pertinent findings.

## 3. Results

### 3.1. Publication Analysis

The search query returned 1081 articles across all three databases, 458 (42%) of which were removed as duplicates. Of the remaining 623 articles, 601 (96%) did not meet inclusion criteria and were excluded. The reference lists and citations of the 22 outstanding articles were then screened, with seven additional articles identified that underwent full text screening. Sixteen articles met full eligibility criteria and were included in the final review. The PRISMA 2020 flow diagram, describing the search strategy and selection schema, is displayed in Figure 2.

Studies were published between 1984 and 2022 (Figure 3). Regarding site location, eight (50%) studies were conducted in the United States [53,54,55,56,57,58,59,60], three (19%) in England [61,62,63], and one (6%) in each of The Netherlands [64], Italy [65], Belgium [66], Germany [67], and Japan [68].

Table 3 presents details on the publication year, geographic location, study design, study population (clinical diagnosis, number of patients, sex, and age), breathwork technique (style and methods), and breathwork protocol (duration of sessions, frequency, treatment period, and follow-up). Table 4 reports outcome measures, including objective, subjective, and custom measures, together with the primary findings of each study.

### 3.2. Participants

All study participants were adults, with a mean age per group of 31.1 ± 9.1 [68] to 45.7 ± 12.5 [58]. Treatment group sizes ranged from four [59] to 92 [66] participants (Table 3). Of the studies included, 15 (94%) had a higher proportion of females, and one (6%) study showed an equal distribution between both males and female sexes [59]. All studies recruited patients clinically diagnosed with an anxiety disorder, in line with the DSM-5 classification system. PD, whether associated with agoraphobia or not, was the most widely investigated condition (*n* = 13, 81%). Apart from this, one (6%) study involved patients with agoraphobia [61], one (6%) with GAD [65], and one (6%) study with acute or chronic hyperventilation syndrome [66] (Table 3).

### 3.3. Study Designs

Seven (44%) studies were randomized controlled trials (RCTs) [55,57,58,62,63,64,67], and nine (56%) were non-randomized, open pilot studies [53,54,56,59,60,61,65,66,68] (Figure 4 and Table 3). Most studies (*n* = 12, 75%) were between subjects, including a control or comparison group [53,54,55,56,57,58,61,62,63,64,67,68]. Only four (25%) studies were within subjects, with no blinding or masking implemented [59,60,65,66] (Table 3). The environments where breathwork interventions took place varied from clinical [61,63,64,67,68] to research [53,54,55,56,57,58,59,62,66] to real-world [60,65] settings (Table 3).

### 3.4. Breathwork Styles

The style of breathwork and its technique and protocol differed among studies (Table 3). In five (31%) studies, breathwork was based on slow diaphragmatic or deep breathing [53,61,63,66,68], although regimens varied (Table 3). Bonn et al. applied diaphragmatic breathing via guided respiration at 8–10 beats per minute (BPM) [61]. Ito et al.’s protocol included slow breathing and self-exposure [63], while Yamada et al. leveraged supine relaxed breathing (using a weight or hand pressure to control diaphragmatic breathing) and seated breathing (with lumbar spines flexed at expiration and dorsiflexed at inspiration) [68]. Conrad et al. implemented different breathwork regimens among three treatment groups, namely, breathing at a slower and/or shallower pace than usual or paying attention to breathing [53].

Three (19%) studies combined different styles of breathwork [62,64,66]. In Han et al., breathing retraining consisted of abdominal breathing with slowed expiration, pre-treated with three minutes of hyperventilation and reattribution of panic symptoms to hyperventilation [66]. Similarly, Hibbert and Chan implemented breathwork that included two sessions, an initial provocation test based on rebreathing using paper bag-forced ventilation, followed by paced or controlled breathing [62]. One study applied breathing retraining cognitive restructuring (BRCR), where slow diaphragmatic breathing was combined with voluntary hyperventilation, catastrophic thinking explanation, and muscle relaxation training [64]. Four (25%) studies included respiratory (or capnometry)- biofeedback-assisted therapy [56,57,59,60], three of which were conducted by Meuret et al. in the United States [56,57,59] (Table 3). Here the breathwork protocols included a two-minute baseline recording, followed by 10 min of paced breathing via recorded tones, reaching BPM targets of 13, 11, 9, and 6 each week, and ending with five minutes of breathing without pacing tones. Herhaus et al. utilized heart rate variability-biofeedback (HRV-BF) training, based on the physiological link between heart rate and breathing, to reach high cardiac coherence (0.1 Hz) [67]. Heart rate variability (HRV) is simply defined as the variance in time between each heartbeat.

Two (13%) studies compared hypercapnic with hypocapnic breathing therapies, where patients were instructed to breathe deeply or shallowly to obtain high or low pCO_2_ [55,58]. In one of these studies, the anxiogenic effects of CO_2_ and hyperventilation were assessed through combined 5% CO_2_ air breathing and 7% CO_2_ air inhalation and hyperventilation [54].

While Sudarshan kriya yoga (SKY) has been widely implemented in studies to manage anxiety, only one study fit our selection criteria [65]. This intervention combines different breathwork styles performed in sequence: Ujjayi, slow breathing; Nadi Shodhana, alternate nostril breathing; Kapalabhati, fast diaphragmatic breathing; Bhastrika, rapid exhalation; and Sudarshan Kriya, rhythmic, cyclical breathing in slow, medium, and fast cycles.

### 3.5. Breathwork Regimens

Breathwork duration and frequency were heterogenous across studies. In particular, interventions ranged from 17 min [55,60] to three hours [53] and a single session [53,54] to 56 sessions delivered over one month [60]. Some studies included follow-up periods of varying lengths and frequencies, spanning a single follow-up assessment one month [58] or two months [59] post-treatment to multiple follow-ups at two, six, and 12 months [60] (Table 3).

### 3.6. Breathwork Outcomes

Outcome measures also differed among studies in the review, which were grouped into objective (based on impartial and quantifiable data), subjective (based on human judgment and experience), and custom (personalized to specific study aims) measures (Table 4). Notably, slow diaphragmatic, deep, or controlled breathing regimens revealed a significant reduction in anxiety in five (31%) studies [61,62,63,66,68] (Figure 5). One study specifically showed marked improvement in the frequency of panic attacks as well as other psychophysiological scores in patients diagnosed with AG compared with controls, which was maintained at the six-month follow-up [61]. Other studies demonstrated significant improvement in panic attack frequency and severity [62], panic in patients with PD and severe AG [63], daily life complaints and state anxiety [66], and percent vital capacity (%VC) and diaphragmatic breathing in PD [68]. Although there was a robust reduction in overall anxiety in both treatment groups, in two of the five aforementioned studies, the two groups (hyperventilators versus non-hyperventilators [62] and self-exposure to internal and external cues versus external cues only [63]) did not differ significantly in anxiety symptoms between them.

All four studies that included respiratory (or capnometry) biofeedback-assisted therapy [56,57,59,60] showed significant improvement in PD severity and measurements (e.g., pCO_2_), maintained at the 12-month follow-up [57,60] (Figure 5). In two of these studies with controls, marked reductions in all measures occurred only in the capnometry-assisted respiratory training group [56,57]. This included corrections from hypocapnic to normocapnic levels [56]. Similarly, Herhaus et al. found that PD or agoraphobia patients receiving HRV-BF training significantly improved their HRV and panic symptoms compared with controls [67]. Given the physiological connection between the breath and the heart, both HRV-BF and CART (capnometry-assisted respiratory training) showed positive outcomes in PD [56,57,59,60,67].

The effects of hyperventilation on PD were assessed in three (19%) studies [54,55,58]. Two studies compared the effects of hypoventilation with hyperventilation by lowering or increasing pCO_2_, respectively. However, the findings were contradictory [55,58]. Kim et al. found that both breathing methods effectively reduced the severity of PD, sustained at the six-month follow-up. Patients also learned how to alter their pCO_2_ and respiratory rate (RR) [55]. In contrast, Wollburg et al. [58] showed that despite hypercapnic and hypocapnic breathing associated with higher and lower baseline pCO_2_ levels, this did not correspond to changes in RR or VT.

One study compared the sensitivity to anxiogenic effects of CO_2_ between panic patients and healthy individuals by assessing the inhalation of 5% CO_2_ and 7% CO_2_ and room-air hyperventilation. They found that room-air hyperventilation caused panic attacks in fewer patients and that CO_2_ was a more potent anxiogenic stimulus, with 7% CO_2_ discriminating best [54].

Conrad et al. showed that paying attention to breathing significantly reduced respiratory and autonomic measures more than anti-hyperventilation instructions. Breathing more slowly, shallowly, or both failed to raise end-tidal pCO_2_ above initial baseline levels for any of the groups [53]. Breathing retraining cognitive restructuring (BRCR), which uses a combination of hyperventilation and slow breathing, led to robust improvement in symptomatology on all self-report measures, except panic frequency and lowered RR [64] (Figure 5). Two (13%) studies demonstrated significant increases in both anxiety and depression [59,65] by implementing SKY [65] or respiratory biofeedback-assisted therapy training [59] (Figure 5).

## 4. Discussion

The link between breathing and anxiety disorders is well established in the field, supported by decades of research on respiratory symptoms [69,70] and hypersensitivity to CO_2_ [71,72]. This association is bidirectional, as observed in patients with asthma [73] or chronic obstructive pulmonary disease (COPD) [74]. Individuals with respiratory dysfunction are also more susceptible to anxiety and depression [73,74]. For instance, comorbid anxiety and depression is an independent predictor of the future risk of asthma [73,75,76,77]. Therefore, targeting respiratory abnormalities through breathwork in patients with anxiety disorders may directly improve physiological, psychological, and behavioral outcomes.

Despite the strong relationship between breathing practices and symptom improvement in anxiety disorders, breathing is not targeted in gold standard treatments. As such, the aim of this scoping review was to characterize the regimens and efficacy of breathwork used among this population. The final review included sixteen studies that implemented a range of breathwork interventions for anxiety (Table 3). Overall, breathwork positively influenced patient outcomes; however, there was high variance in breathwork styles and protocols, and in a few cases, the results were contradictory [55,58]. Notwithstanding, all studies utilizing slow diaphragmatic breathing showed positive effects on stress reduction as well as significant improvement in outcome measures (Table 4). The effectiveness of diaphragmatic breathing for decreasing stress in healthy adults has been reviewed elsewhere [38].

Biofeedback (BF), a non-invasive and interactive mind–body technique, where individuals learn how to change their physiological activity, is commonly used for stress management [78]. The technique is relatively quick and inexpensive and can be implemented in both real-world and clinical settings, as demonstrated in studies included in this review. While protocols varied in terms of duration, number of sessions, and follow-up, studies outside of this review have reliably shown benefits for anxiety, depression, and cognition even after a single session [79,80] and in as little as five minutes [81,82].

By interacting with the cardiorespiratory system via increased HRV and respiratory sinus arrhythmia (RSA), slow breathing techniques augment parasympathetic versus sympathetic activity, which impacts emotional regulation and well-being [38,83] (Figure 1). The cardiac vagal tone is measured indirectly through HRV, with lower frequencies associated with a number of psychopathological conditions, including anxiety and depression, and is a predictor of mortality [84,85]. Hence, HRV-BF has received increasing attention for improving health, mood, and stress, carrying therapeutic potential for anxiety disorders [86,87]. Several possible mechanisms for the effectiveness of HRV-BF have been proposed [88]. Operationally, HRV-BF is connected to paced or coherent breathing at 0.1 Hz (six breaths per minute). This maximizes HRV and synchronizes pulse harmonics of blood flow with heart rhythms [67,89]. HRV is therefore directly influenced by breathing and RSA, or the fluctuation in heart rate relative to breathing, increasing with inhalation and decreasing with exhalation. Accordingly, RSA is a marker of cardiac-linked parasympathetic and emotional regulation as well as cognition [90]. This aligns with our present analysis, showing that both HRV-BF and capnometry-guided respiratory training are efficacious for anxiety in clinically diagnosed patients [56,57,59,60,67], as they are for relapse prevention over time (2 to 12 months) [57,59,60] (Figure 5).

In synthesizing the considerations above, we conclude that breathwork interventions, such as diaphragmatic breathing and respiratory or HRV-assisted therapies, widely used to reduce stress among the general population [38,88,91], are effective in targeting panic and stress in patients clinically diagnosed with anxiety. However, our analysis also suggests that breathwork techniques and protocols play significant roles in health outcomes. One study that instructed patients to breathe more slowly and/or shallowly did not produce changes in respiratory or autonomic measures toward relaxation [53]. Yet, the same study showed that paying attention to breathing, commonly recommended in practices such as yoga and meditation, led to significant improvements in physiological measures. They cautioned interpreting these findings due to their small sample size (*n* = 13), which was not informed by a prior pilot study. Additionally, investigators were unable to separate medicated versus non-medicated PD patients, despite several of them receiving relaxation agents [53].

The most controversial findings elucidated in this review were hyperventilation inducing anxiogenic effects [54] or improving panic symptoms in PD [55]. This reflects two inconclusive theories of PD, regarding the causal relationship between respiratory regulation (pCO_2_) and PD: Ley’s hyperventilation theory [92] and Klein’s suffocation false alarm theory [93]. In Ley’s theory, hyperventilation plays a causative role in the onset of panic attacks and often follows beyond the patient’s awareness [92]. This theory is supported by evidence that voluntary hyperventilation increases anxiety in patients with PD and can trigger panic attacks [94]. Supporting this theory, one study demonstrated that patients with PD were more sensitive to the anxiogenic effects of CO_2_ than controls, which was a more potent anxiogenic stimulus compared with room-air hyperventilation. [54]. In Klein’s suffocation theory, hyperventilation is viewed not as a cause of panic but rather as a compensatory adaptation in reaction to an oversensitive “suffocation alarm system”, keeping pCO_2_ sufficiently low to avoid triggering the suffocation alarm [93]. This theory was supported by two studies in our review [55,64].

In the first study, de Ruiter et al. concluded that hyperventilation plays a less important role in panic than previously thought [64]. A reinterpretation of experimental findings by de Ruiter et al. was performed by Ley in 1991. The authors pointed to the complications and complexity of breathing retraining; for instance, hypoventilation through reduction in RR paradoxically increases ventilation [95]. In the second study, Kim et al. showed that two opposing retraining protocols (raising versus lowering end tidal CO_2_) produce equivalent therapeutic outcomes for panic or episodic anxiety attacks, specifically by reducing panic attacks for six months post-treatment [55]. Investigators suggested that both breathing therapies should be implemented in the clinic for the treatment of PD. The results of this study align with others included in this review, wherein positive outcomes of PD are observed when interventions include voluntary hyperventilation combined with slow, deep, or diaphragmatic breathing [62,64,66]. However, a commentary on the study published by Kim et al. points to limitations related to the experimental design and the small sample size of the study and suggests that breathing retraining aimed at raising pCO_2_ may be supported in panic patients [96].

Other studies have suggested that acute hyperventilation is neither necessary nor sufficient for panic to occur and that patients with PD might implement adaptive and/or abnormal breathing to cope with a hypersensitive CO_2_ chemoreceptor system [72]. Regardless, breathing retraining for PD is supported by both theories, with opposite respiratory goals. If the hyperventilation theory is true, breathwork should aim to prevent hyperventilation and subsequent panic attacks. If the false-suffocation alarm theory is true, then lowering pCO_2_—below the threshold for triggering the suffocation alarm—should be used instead.

In addition to Ley’s and Klein’s theories, we propose a third interpretation for the role of hyperventilation in PD. Holotropic breathwork, translating to “moving towards wholeness” [97], which combines faster and deeper breathing to induce intense altered states of consciousness, has been shown to increase self-awareness [98]. It also has been proposed as an adjunct to psychotherapy to facilitate generalized extinction of avoidance behaviors that could be useful in the treatment of anxiety and depressive disorders [47]. Although the potential benefits of holotropic breathwork in anxiety disorders are yet to be investigated, we hypothesize that the beneficial mechanisms of hyperventilation in breathing retraining, examined here, are common to holotropic breathwork and other techniques that invoke non-ordinary states of consciousness.

### Limitations

The limitations of this scoping review are related to breathwork heterogeneity, namely, variance in interventional styles and protocols (Table 3). Across studies, there was lack of homogeneity in patient populations and outcome measures. Three studies did not include objective measures [62,63,65], and others did not include a follow-up assessment [53,54,56,62,64,66,67,68]. Sample size was an additional limitation present in most studies [53,54,55,58,59,61,62,63,64,67,68], as was the lack of a comparison or control treatment [59,60,65,66]. Finally, few studies did not involve randomization [53,54,56,59,60,61,65,66,68], and others did not include a medication washout [53,62,63,64,65,66].

## 5. Conclusions

This scoping review supports the clinical utility of breathwork interventions and discusses effective treatment options and protocols that are feasible and accessible to patients suffering anxiety. Future research directions include the optimization and standardization of breathwork practices and protocols for the treatment of anxiety disorders.

## Figures and Tables

**Figure 1 brainsci-13-00256-f001:**
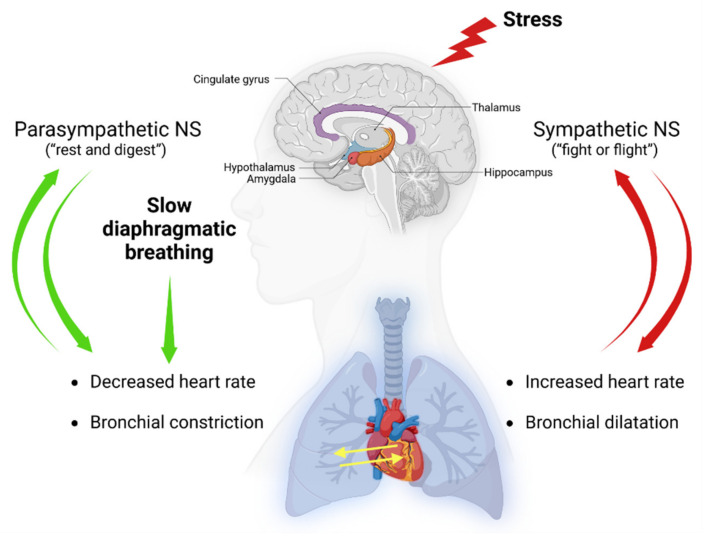
Effects of slow-diaphragmatic breathwork practices on the nervous system. Slow diaphragmatic breathing has an impact on the cardiovascular and respiratory systems and can stimulate the parasympathetic activity of the nervous system (NS). Stress causes hyperactivation of the sympathetic nervous system and results in cardioacceleration and bronchial dilatation. Generated using Biorender, https://biorender.com/, accessed on 17 December 2022.

**Figure 2 brainsci-13-00256-f002:**
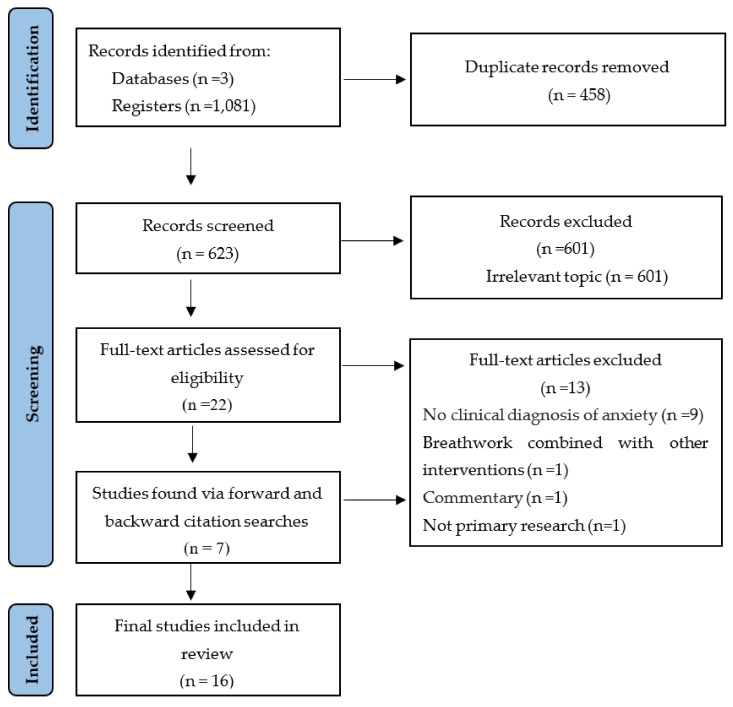
PRISMA flowchart of the study selection process.

**Figure 3 brainsci-13-00256-f003:**
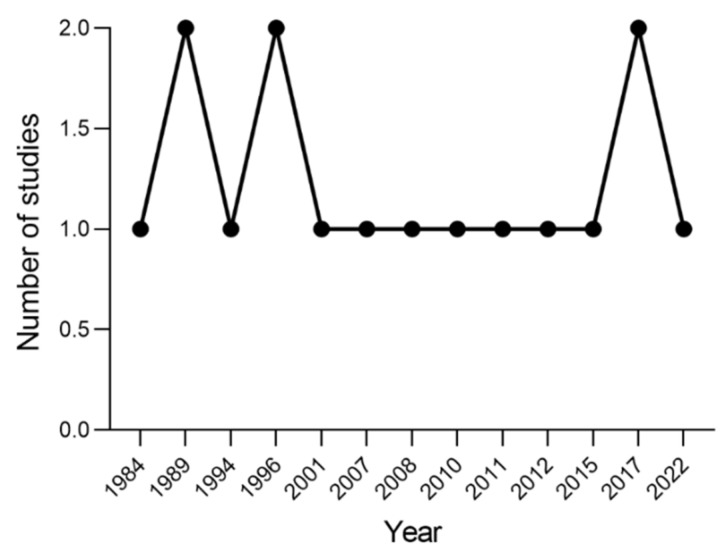
Number of studies published per year.

**Figure 4 brainsci-13-00256-f004:**
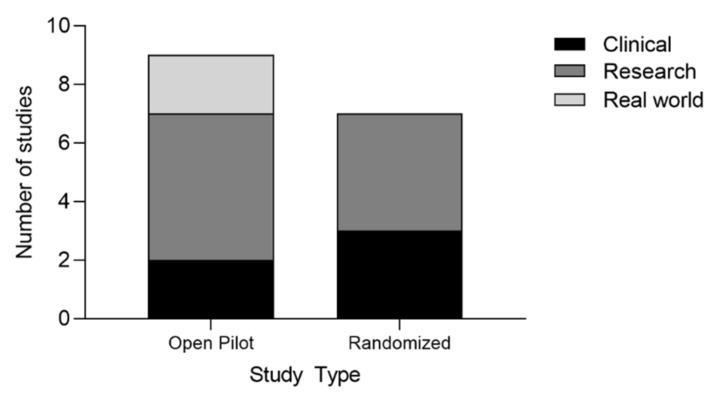
Number of studies per type and setting.

**Figure 5 brainsci-13-00256-f005:**
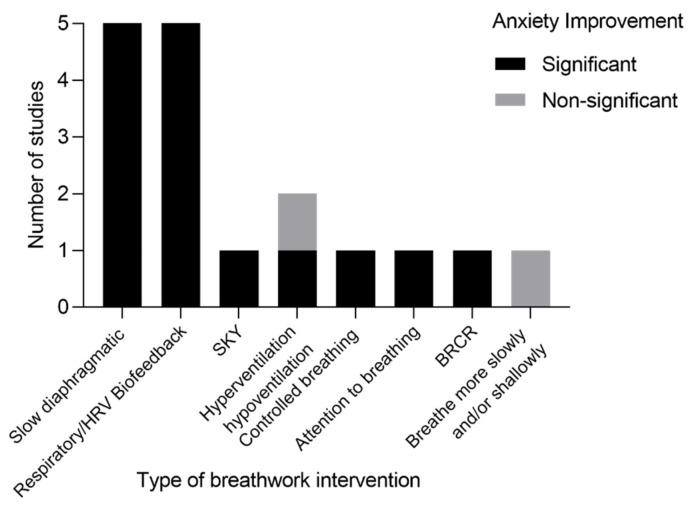
Number of studies per style of breathwork with relative outcome on anxiety improvement (significant or not significant). Abbreviations: breathing retraining cognitive restructuring (BRCR), Surdashan Kriya Yoga (SKY), and heart rate variability-biofeedback (HRV-BF).

**Table 1 brainsci-13-00256-t001:** Search terms.

**Breathwork search terms**	*Breathwork, *breathing exercise, *breathing technique, *breathing practice, *breath regulation, *pranayama, *mindful breathing, *paced breathing, *controlled breathing, *slow breathing, *fast breathing, *hyperventilation, *deep breathing, *metronome breathing, *nasal breathing, *mouth breathing, *diaphragmatic breathing
**Anxiety search terms** (DSM-5 classification system)	*Anxiety, *anxiety disorders, *stress disorders, *stress, *anxiety, *phobia, *phobic, *panic, *stress disorder, *agoraphobia

**Table 2 brainsci-13-00256-t002:** Inclusion and exclusion criteria via PICOS.

PICOS Criteria	Inclusion Criteria	Exclusion Criteria
**P (population)**	Patients diagnosed with an anxiety or stress-related disorder using the DSM-5 classification system Patients aged >18 years (adults) Patients of any biological sex, ethno-racial, and gender identity	Patients not diagnosed with an anxiety or stress-related disorder according to the DSM-5 classification system Patients aged < 18 years (children and adolescents)
**I (intervention)**	Breathwork interventions or techniques or exercises	Non-breathwork interventions or techniques or exercises Breathwork interventions or techniques or exercises combined with movement (e.g., yoga or Pilates) or techniques (meditation or visualization) Breathwork interventions or techniques or exercises combined with other interventions (dietary changes or supplementation, exercise, etc.)
**O (outcome)**	Study year, study design, study source/location, condition, number of patients, intervention type, intervention regimen, measures, cognitive outcomes, behavioral outcomes, and quality of life outcomes	Not applicable
**S (study design)**	Clinical trials Controlled clinical trials Non-randomized or randomized clinical trials Articles published in the English language	Articles that do not include clinical trials, controlled clinical trials, or non-randomized or randomized clinical trials Student dissertations or theses Conference abstracts or articles Methodology, protocol, or technical note papers Reviews (narrative, systematic, scoping, or meta-analysis) Editorials, commentaries, or opinion pieces Case reports or case series Articles without full texts Articles not published in the English language

**Table 3 brainsci-13-00256-t003:** Intervention details.

		Population Characteristics			Breathwork Intervention Characteristics			
**Study reference/** **year/** **country**	**Study design** a. Study type b. Design c. Setting	**Diagnosis**	**Experimental group** a. No. of participants (N) and sex b. Mean age	**Control group** a. No. of participants (N) and sex b. Mean age	**Type of breathwork intervention**	**Breathwork protocol** **(treatment)**	**Details of breathwork intervention** a. Duration of session b. No. of sessions c. Treatment duration d. Follow-up	**Breathwork protocol** **(control)**
Bonn et al. [61] 1984 England	a. Open pilot b. Between subject c. Clinical	Agoraphobia	a. N = 7 (4 females, 3 males) b. 35.5	a. N = 5 (3 females, 2 males) b. 39	Slow diaphragmatic breathing	Breathing retraining via guided diaphragmatic respiration, 8–10 BPM	a. 120 min b. 9 sessions c. 10 weeks d. 1, 6 months	Real-life exposure (9 × 120 min/9 weeks)
Conrad et al. [53] 2007 United States	a. Open pilot b. Between subject c. Research	Panic disorder	a. N = 13 (11 females, 2 males) b. 39.2	a. N = 15 (10 females, 5 males) b. 40.7	Direct attention to breathing, instructions to breathe more slowly and/or shallowly	Paying attention to breathing or breathing at a slower pace than usual or breathing shallower than usual or breathing shallower and slower than usual	a. 180 min b. 1 session c. NA d. NA	Same as treatment
de Ruiter et al. [64] 1989 The Netherlands	a. Randomized trial b. Between subject c. Clinical	Panic disorder with agoraphobia	a. N = 13 (7 females, 6 males) b. 34.0	a. N = 27 (17 females, 10 males) b. 34.0	Breathing retraining cognitive restructuring (BRCR)	Voluntary hyperventilation, hyperventilation and catastrophic thinking explanation, muscle relaxation training, slow diaphragmatic breathing training	a. 60 min b. 8 sessions c. NA d. NA	Graded self-exposure in vivo (8 × 60 min); breathing retraining or cognitive restructuring (4 × 60 min) plus graded self-exposure in vivo (4 × 60 min)
Gorman et al. [54] 1994 United States	a. Open pilot b. Between subject c. Research	Panic disorder/ agoraphobia	a. N = 24 (15 females, 9 males) b. 36.3	a. N = 18 (12 females, 6 males) b. 32.5	CO_2_ inhalation and room-air hyperventilation	(a) Air breathing (20 min). (b) 5% CO_2_ (20 min) or hyperventilation (15 min), metronome-guided breathing (30 BPM). (c) Air breathing (15 min). (d) 5% CO_2_ (20 min) or hyperventilation (15 min). (e) Air breathing (15 min). (f) 7% CO_2_ (20 min).	a. 35 min b. 1 session c. NA d. NA	Same as treatment
Herhaus et al. [67] 2022 Germany	a. Randomized trial b. Between subject c. Clinical	Panic disorder/ agoraphobia	a. N = 26 (19 females, 7 males) b. 35.7	a. N = 26 (17 females, 9 males) b. 35.7	Heart rate variability-biofeedback (HRV-BF) training	Heart rate variability-biofeedback at 0.1 Hz breathing (10 × 20 min), and at-home breathing training with 0.1 Hz breathing via audio track (2 × 20 min/day)	a. 20 min b. 10 sessions c. 4 weeks d. NA	Sham HRV-B biofeedback without paced breathing (10 × 20 min/4 weeks)
Hibbert and Chan [62] 1989 England	a. Randomized trial b. Between subject c. Research	Panic disorder with agoraphobia	a. N = 21 (13 females, 8 males) b. 35.0	a. N = 19 (13 females, 6 males) b. 35	Controlled breathing training	First treatment session: provocation test (rebreathing using a paper bag, 2 min of forced ventilation, rebreathing with the bag); second treatment session: paced breathing, overbreathing	a. Not reported b. 2 sessions c. 3 weeks d. NA	Psychoeducation (1 ×/1 week) plus diary homework (1 × 30 min/day) followed by anxiety management strategies
Ito et al. [63] 1996 England	a. Randomized trial b. Between subject c. Clinical	Panic disorder with agoraphobia	a. N = 10 b. 37.0	a. N = 9 b. 38	Slow deep breathing	Slowed breathing (2 × 5 min/day) plus self-exposure (30–45 min) × 7 sessions/10 weeks, plus therapist-accompanied internal exposure (2 × 30 min/2 weeks). Daily exposure homework: 60 min external plus 30 min internal exposure. Exposure protocol: hyperventilation (1 min), shaking head side to side (30 s), running on spot (1 min), head between legs (30 s) and then quickly lifting head up. Slow breathing after each exercise to return to baseline anxiety level.	a. Not reported b. 7 sessions c. 10 weeks d. 4, 10, 14, 24 weeks	Slowed breathing (2× 5 min/day) plus self-exposure (30–45 min) × 7 sessions/10 weeks, plus therapist-accompanied external exposure (1 × 60 min/1 week). Daily between-session exposure homework: 90 min of external exposure. Exposure protocol same as treatment.
Kim et al. [55] 2012 United States	a. Randomized trial b. Between subject c. Research	Panic disorder/ agoraphobia	a. N = 74 (51 females, 23 males) b. 41.9	a. N = 30 (21 females, 9 males) b. 43	Hypoventilation, hyperventilation	Biofeedback-assisted hypercapnic or hypocapnic breathing therapy (5 ×/5 weeks) plus at-home breathing exercises (2 × day/5 weeks). Breathing exercises: baseline breathing (2 min), breathing more deeply or shallowly to reach PCO_2_ = 30 mm Hg (lower-CO_2_ group) or 40 mm Hg (raise-CO_2_ group) by breathing via audio tape (10 min) followed by breathing without audio tape (5 min). RR maintained at 9 BPM.	a. 17 min b. 5 sessions c. 5 weeks d. 1, 6 months	Delayed treatment
Meuret et al. [56] 2010 United States	a. Open pilot b. Between subject c. Research	Panic disorder with agoraphobia	a. N = 21 (17 females, 4 males) b. 31.4	a. N = 20 (17 females, 3 males) b. 35	Capnometry-assisted respiratory training (CART)	Education and teaching to control end-tidal PCO_2_ and RR. Between-session exercises (2 × 17 min/day): (a) 2 min physiological baseline recording. (b) 10 min synchronized breathing with recorded tones (weekly BPM targets: 13, 11, 9, 6) and PCO_2_ of 40 ± 3 mm Hg. (c) 5 min transfer with visual feedback.	a. 60 min b. 5 sessions c. 4 weeks d. None	Cognitive skill training (5 × 60 min/week) plus cognitive homework (2 × 17 min/day)
Meuret et al. [57] 2008 United States	a. Randomized trial b. Between subject c. Research	Panic disorder/ agoraphobia	a. N = 20 b. 41.0	a. N = 17 b. 41	Capnometry-assisted breathing therapy (BRT)	Education, direct attention to respiratory patterns, breathing maneuvers, teaching control of pCO_2_ and RR. Daily breathing exercises (2 × 17 min/day): (a) 2 min physiological baseline recording. (b) 10 min synchronized breathing with recorded tones (weekly BPM targets: 13, 11, 9, 6) in a normocapnic range (pCO_2_ > 35 mmHg. (c) 5 min transfer, breathing without pacing tones.	a. 60 min b. 5 sessions c. 4 weeks d. 2, 12 months	Delayed treatment
Wollburg et al. [58] 2011 United States	a. Randomized trial b. Between subject c. Research	Panic disorder	a. N = 45 (30 females, 15 males) b. 41.8	a. N = 20 (11 females, 9 males) b. 45.7	Hypoventilation, hyperventilation	(a) Education. (b) Biofeedback-assisted hypercapnic or hypocapnic breathing therapy (5 ×/5 weeks). (c) At-home breathing exercises (2 × day/5 weeks): training by breathing more deeply or shallowly to obtain the target pCO_2_ (lower: 30 mmHg, raise: 40 mmHg) at 9 BPM.	a. Not reported b. 5 sessions c. NA d. 1 month	Delayed treatment
Yamada et al. [68] 2017 Japan	a. Open pilot b. Between subject c. Clinical	Panic disorder	a. N = 28 (20 females, 8 males) b. 31.1	a. N = 28 (20 females, 8 males) b. 31.1	Slow diaphragmatic breathing	CBT (weekly). Diaphragmatic breathing retraining (daily): (a) relaxed breathing–supine breathing training w/500 g weight and hand pressure. (b) Seated breathing training w/lumbar spine flexed w/expiration and dorsiflexed w/inspiration. Stretching exercises for breathing muscles.	a. Not reported b. CBT weekly, BT daily. c. 6–13 weeks d. None	Same as treatment
Doria et al. [65] 2015 Italy	a. Open pilot b. Within subject c. Real world	Generalized anxiety disorder	a. N = 69 (41 females, 28 males) b. Not specified	NA	Surdashan Kriya Yoga (SKY)	Ujjayi, slow breathing 3–4 cycles per minute; Nadi Shodhana, alternate nostril breathing, Kapalabhati, fast diaphragmatic breathing; Bhastrika, rapid exhalation at 20–30 cycles/min; and Sudarshan Kriya, rhythmic, cyclical breathing in slow, medium, and fast cycles	a. 120 min b. 10 sessions c. 2 weeks d. 2 weeks, 3 and 6 months	NA
Han et al. [66] 1996 Belgium	a. Open pilot b. Within subject c. Research	Hyperventilation syndrome	a. N = 92 (60 females, 32 males) b. 37.0	NA	Abdominal slow breathing	Hyperventilation (3 min), reattribution of symptoms to hyperventilation, abdominal breathing with slowed expiration, breathing retraining	a. 45 min b. 17 sessions c. 2.5 months d. NA	NA
Meuret et al. [59] 2001 United States	a. Open pilot b. Within subject c. Research	Panic disorder/ agoraphobia	a. N = 4 (2 females, 2 males) b. 42.0	NA	Respiratory biofeedback-assisted therapy	Education, teaching techniques to control respirations, direct attention to respiratory patterns. Home breathing exercises (2 × 17 min/day): (a) 2 min baseline recording. (b) 10 min paced breathing with recorded tones (weekly BPM targets: 13, 11, 9, 6). (c) 5 min transfer, breathing without pacing tones.	a. 80 min b. 5 sessions c. 4 weeks d. 2 months	NA
Tolin et al. [60] 2017 United States	a. Open pilot b. Within subject c. Real world	Panic disorder	a. N = 69 (41 females, 28 males) b. 36.6	NA	Capnometry guided respiratory intervention (CGRI)	Breathing sessions (2 × 17 min/day): (a) baseline breathing (2 min), (b) 10 min paced breathing via audio tape (weekly BPM targets: 13, 11, 9, 6), target PETCO_2_ level (37–40 mmHg). (c) 5 min transfer, breathing without pacing tones.	a. 17 min b. 56 sessions c. 4 weeks d. 2, 6, 12 months	NA

**Table 4 brainsci-13-00256-t004:** Study measures and outcomes.

	Study Measures			
Study Reference	Objective Measures	Subjective Measures	Custom Measures	Primary Findings
Bonn et al. [61]	Panic attack frequency, resting breathing rate	NA	Phobia and agoraphobia scores, somatic symptoms	Significant improvement in all measures compared with control, maintained at 6-month follow-up
Conrad et al. [53]	Skin conductance, pCO_2_, tidal volume, HR	ASI, BAI, BDI, PSS, PSWQ	Mood questionnaire	No change in respiratory or autonomic measures in direction of relaxation, except for attention to breathing
de Ruiter et al. [64]	pCO_2_, RR	FBSQ, FSS-III, SCL-90	Phobic anxiety and avoidance scales, panic attack diary	Significant improvement in all measures, except panic frequency, with reduction in RR; no significant differences between groups
Gorman et al. [54]	pCO_2_	API, RPE, SADS-LA	Anxiety and apprehension scales	Significant sensitivity to anxiogenic effects of CO_2_ compared with controls; 7% CO_2_ discriminated best. CO_2_ is a more potent anxiogenic stimulus than room-air hyperventilation.
Herhaus et al. [67]	HR, BMP, EEG signals	ACQ, ASI, BDI, BSQ, MI, PAS	NA	Improved HRV and panic symptoms compared with controls
Hibbert and Chan [62]	NA	BDI, FQ, HARS	Panic attack and exposure diary	Significant improvement in anxiety measures compared with controls; no significant differences in hyperventilators compared with non-hyperventilators
Ito et al. [63]	NA	ACQ, BDI, BSQ, FQ, HARS, PT, disability (measure not defined)	Panic attack diary	Significant improvement in all measures at post-treatment and follow-up; no significant differences between groups, but a slightly greater number of patients in the treatment group showed improvement in phobic avoidance and fear
Kim et al. [55]	pCO_2_, RR	ACQ, ACQ, ASI, BAI, BDI, MIA, PDSS	NA	Significant improvement in panic severity by using both breathing training methods, maintained at 6-month follow-up; patients learned to alter pCO_2_ and RR via therapy
Meuret et al. [56]	pCO_2_, RR	BSQ, ASI, BSQ/ASI, ACQ, PDSS, CEQ	NA	Significant improvement in panic symptoms, panic-related cognitions, and perceived control in both treatment groups; corrections from hypocapnic to normocapnic levels only in capnometry-assisted respiratory training group
Meuret et al. [57]	pCO_2_, RR	PDSS, clinician-rated PD severity, CGI, ASI, SDS, MI-AAL, BDI	NA	Significant improvement in all measures in treated but not untreated patients; psychological outcomes maintained at 2- and 12-month follow-up
Wollburg et al. [58]	Expired CO_2_, end-tidal pCO_2_, RR, respiratory rate instability, TV, TV instability, HR, skin conductance	ASI, MI, BDI, ASQ	PDSS, ASI, mobility inventory for agoraphobia, BDI, anxiety symptom checklist, pCO_2_, expired CO_2_, RR, RR instability, tidal volume, TV instability, HR, respiratory sinus arrhythmia, skin conductance	Before treatment: higher respiration rates, tidal volume instability and sighing at rest of panic patients compared with non-anxious controls. After lowering pCO_2_ therapy: lower pCO_2_ during testing of panic patients but no significant differences in reactivity, recovery, or treatment effect between groups; baseline abnormalities somewhat specific to PD
Yamada et al. [68]	VC, %VC	Diaphragmatic breathing assessment	NA	%VC was significantly reduced in patients with impaired diaphragmatic breathing, which was equally recoverable compared with controls, with breathing retraining
Doria et al. [65]	NA	HRSA, HRSD, SCL-90, ZSAS, ZSDS	NA	Significant improvement in anxiety and depression
Han et al. [66]	pCO_2_, inspiratory time, inspiratory volume, expiratory time, expiratory volume	NQ, STAI, ZBV-DY1, ZBV-DY2	NA	Significantly improved daily life complaints and state anxiety; markedly changed breathing pattern
Meuret et al. [59]	pCO_2_, RR	PDSS, ASI, STAIT-T, BDI	Rate average anxiety, depression, anticipation, and worry daily; panic attack diary	Significant improvement in PD symptoms and pCO_2_ that continued through follow-up; equal reductions in fear, anxiety and depression
Tolin et al. [60]	pCO_2_, RR	PDSS, NINI, CGI-S, SDS, MI-AAL, ASI, BDI, panic frequency	Patient satisfaction	Significant improvement in PD severity, with high treatment response and remission maintained at 12-month follow-up. Decrease in functional impairment and global illness severity.

ACQ = Agoraphobic Cognitions Questionnaire; API= Acute Panic Inventory; ASI = Anxiety Sensitivity Index; ASQ = Attributional Style Questionnaire; BAI = Beck Anxiety Inventory; BDI = Beck Depression Inventory; BSQ = Body Sensations Questionnaire; CEQ = Credibility/Expectancy Questionnaire; CGI = Clinician Global Impression; CGI-S = Clinician Global Impression-Severity; FBSQ = Fear of Bodily Sensations Questionnaire; FQ = Fear Questionnaire; FSS-III = Fear Surrey Schedule-III; HARS = Hamilton Anxiety Rating Scale; HR = heart rate; HRS-A = Hamilton Rating Scale for Anxiety; HRS-D = Hamilton Rating Scale for Depression; HRV = heart rate variability; MI = Mobility Inventory; MIA = Mobility Inventory for Agoraphobia; MI-AAL = Mobility Inventory for Agoraphobia; MINl = Mini International Diagnostic Interview; NQ = Nijmegen Questionnaire; PAS = Panic and Agoraphobia Scale; pC02 = carbon dioxide pressure; PD = panic disorder; PDSS = Panic Disorder Severity Scale; PSS = Perceived Stress Scale; PSWQ = Penn State Worry Questionnaire; PT = Phobic Target; RPE = Rating of Perceived Exertion; RR = respiratory rate; SADS-LA = Lifetime Version Modified for the Study of Anxiety Disorders; SCL-90 = Symptom CheckIist-90; SDS = Sheehan Disability Scale; STAI = State-Trait Anxiety Inventory; STAI-T = State-Trait Anxiety Inventory–Trait subscale; ZSAS = Zung Self-Rating Anxiety Scale; ZSDS = Zung Self-Rating Depression Scale; ZBV DYl, ZBV-DY2 = state and trait versions of the Zelfbeoordelingsvragenlijst.

## Data Availability

Data are available from the corresponding author upon request.

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
