# Peer review of "Breathwork Interventions for Adults with Clinically Diagnosed Anxiety Disorders: A Scoping Review"

_brainsci, 2023, doi:10.3390/brainsci13020256_

Round 1
Reviewer 1 Report
Please adjust the flow chart by adding number of research articles in the final selection.
Can the authors more discuss as the concerning points?
why the author did not mentioned mental disorders with the ICD-11 classifies?
Dusscuss breathing e.g., low Co2 or hyperventilation associated with nervous system (stress-induce hyperactivity of sympathetic nervous system)
Author Response
Please adjust the flow chart by adding number of research articles in the final selection.
Corrected
Can the authors more discuss as the concerning points? why the author did not mention mental disorders with the ICD-11 classifies?
We thank the reviewer for this question. This review is specifically focused on anxiety disorders and no other mental disorders. From a recent publication comparing the organization and diagnostic requirements for mental disorders in ICD‐11 and DSM‐5, the organization of the two classifications of mental disorders was shown to be substantially similar and ‘differences are largely based on the differing priorities and uses of the two diagnostic systems and on differing interpretations of the evidence’ [1].
We based our search terms on the DSM‐5 system. The different diagnosis criteria used by the two systems go beyond the scope of this review and are not relevant to the search strategy.
- Fernández, A. Maercker, C. R. Brewin, M. Cloitre, A. Claudino, K. M. Pike, G. Baird, D. Skuse, R. B. Krueger, P. Briken, J. D. Burke, J. E. Lochman, S. C. Evans, D. W. Woods, and G. M. Reed. "An Organization- and Category-Level Comparison of Diagnostic Requirements for Mental Disorders in Icd-11 and Dsm-5." World Psychiatry 20, no. 1 (2021): 34-51.
Discuss breathing e.g., low Co2 or hyperventilation associated with nervous system (stress-induce hyperactivity of sympathetic nervous system)
We thank the Reviewer for this comment and the opportunity to explain better the association of breathing with the nervous system. To address this point in response to the reviewer 1 and 3, we changed our graphical abstract to Figure 1, with some modifications, and further discussed these concepts in the main text (lines 111-136). We believe this substantially improves the understanding of the association of breathing with the nervous system.
Reviewer 2 Report
This evidence-based review confirms the clinical benefits of breathing interventions and offers effective treatment options and protocols that are feasible and accessible for patients suffering from anxiety. Existing knowledge gaps for future research will be interested for the Brain Sciences readers
The goal of review is to understand efficacy of breathwork interventions for adults with clinically diagnosed anxiety disorders
The topic is original and there are no recent reviews in this field.
This is the first review of breathing techniques in clinically diagnosed anxiety.
There is no comments about methodology.
Conclusions consistent with presented results that breathwork interventions
such as diaphragmatic breathing and respiratory or HRV-assisted therapies, widely used
to reduce stress among the general population are effective in targeting panic and
stress in patients clinically diagnosed with anxiety.
The references are appropriate. There are no comments on tables and figures.
Author Response
We thank the Reviewer for their positive feedback on the review and its clinical implications.
Reviewer 3 Report
The title of the work is adequate, clear and understandable. Its length is relevant.
As a suggestion, it is recommended to delimit or limit the number of keywords to no more than 3-5 at most, thinking about their conceptual and theoretical suitability.
The work has a theoretical character as it is a systematic review of the literature.
The introduction and theoretical contextualization of the work is sufficient. The main variables analyzed are addressed, at least superficially. They provide clear elements regarding anxiety disorders, their characteristics and psychiatric nosology. However, the connection they establish with respiratory abnormalities is not entirely clear. Therefore, it would be desirable if they could clarify this relationship and provide more specific elements about the research problem posed. In the same way, the explanation they provide about dysfunctional breathing in people suffering from anxiety disorders can be improved.
Regarding the method, it is a systematic literature review using the PRISMA method. They provide specific elements about the way in which they oriented the search, the selected databases and the proper compliance with the PRISMA checklist. However, it is not clear why they did not include articles indexed in the Web of Science. Similarly, one of the limitations of the work is that they did not register the search protocol on the PROSPERO platform. On the other hand, the search words were quite broad and the exclusion criteria were very weak. It is recommended to review and improve this section.
The presentation of results is adequate and precise. However, the tables can be improved in their content and explanation/interpretation of the collected findings. The graphics used are adequately achieved.
The discussion is constructed in a pertinent manner, seeking an adequate analysis of the main findings and their relationship with recent studies in the thematic field addressed. However, a greater level of depth would be desirable in relation to the clinical, psychosocial and/or physiological implications related to respiratory work interventions in people and risk groups prone to manifesting anxious symptoms. In other words, it is about providing more robust elements about its therapeutic efficacy depending on the type of anxiety disorder presented. On the other hand, a theoretical improvement regarding acute hyperventilation and its implications in the treatment of anxiety in people with a chronic course of the disease would be desirable.
The manuscript does not present a conclusions section. It is recommended to incorporate a brief, explicit and forceful section that gives a synthetic answer to the purpose of the study.
The limitations of the study are adequate.
A thorough and systematic review of the references section is recommended, trying to comply with the editorial standards of the journal.
Author Response
The title of the work is adequate, clear and understandable. Its length is relevant. As a suggestion, it is recommended to delimit or limit the number of keywords to no more than 3-5 at most, thinking about their conceptual and theoretical suitability. The work has a theoretical character as it is a systematic review of the literature.
We thank the Reviewer for their positive comments.
The number of keywords utilised is within the guidelines of the journal. All keywords utilised are relevant to the study and were adopted in the search strategy of this review.
The introduction and theoretical contextualization of the work is sufficient. The main variables analyzed are addressed, at least superficially. They provide clear elements regarding anxiety disorders, their characteristics and psychiatric nosology. However, the connection they establish with respiratory abnormalities is not entirely clear. Therefore, it would be desirable if they could clarify this relationship and provide more specific elements about the research problem posed. In the same way, the explanation they provide about dysfunctional breathing in people suffering from anxiety disorders can be improved.
We thank the Reviewer for their positive comments on the Introduction. We improved this point also in response to Reviewer 1. See Figure 1 and main text (lines 111-136).
Regarding the method, it is a systematic literature review using the PRISMA method. They provide specific elements about the way in which they oriented the search, the selected databases and the proper compliance with the PRISMA checklist. However, it is not clear why they did not include articles indexed in the Web of Science. Similarly, one of the limitations of the work is that they did not register the search protocol on the PROSPERO platform. On the other hand, the search words were quite broad and the exclusion criteria were very weak. It is recommended to review and improve this section.
We thank the Reviewer for their comments and suggestions on the Methods. We used three large citation databases for our searches: PubMed (National Library of Medicine), Embase (Elsevier), and Scopus (Elsevier). While additional databases could have been used, all authors think that the combinations of the three databases adopted offers an adequate and efficient coverage of all relevant articles.
Our search terms perfectly match the terminology used for breathwork practices and anxiety disorders following the DSM-5 classification system (Table 1).
We think that the exclusion criteria used in this scoping review are very strong because, differently from other systematic reviews, we exclude interventions where breathwork is combined with movement or other non-psychotherapeutic interventions. Additionally, patients not clinically diagnosed with an anxiety or stress-related disorder according to the DSM-5 classification system were excluded, which further strengthens our exclusion criteria. We added a sentence (line 178) in the Methods for further clarification.
PROSPERO does not offer pre-registration of scoping reviews. We pre-registered the review protocol with OSF and added the Identifier in the Methods.
Identifier: osf-registrations-a768w-v1
Osf_project: https://api.osf.io/v2/nodes/kty3s/?version=2.20
Osf_registration: doi 10.17605/OSF.IO/A768W
The presentation of results is adequate and precise. However, the tables can be improved in their content and explanation/interpretation of the collected findings. The graphics used are adequately achieved.
We thank the Reviewer for their positive comments on the Results. We think that the information reported in the tables is exhaustive and standard for this type of review. The interpretation and discussion of the results is reported in the Discussion section of the review.
The discussion is constructed in a pertinent manner, seeking an adequate analysis of the main findings and their relationship with recent studies in the thematic field addressed. However, a greater level of depth would be desirable in relation to the clinical, psychosocial and/or physiological implications related to respiratory work interventions in people and risk groups prone to manifesting anxious symptoms. In other words, it is about providing more robust elements about its therapeutic efficacy depending on the type of anxiety disorder presented. On the other hand, a theoretical improvement regarding acute hyperventilation and its implications in the treatment of anxiety in people with a chronic course of the disease would be desirable. The manuscript does not present a conclusions section. It is recommended to incorporate a brief, explicit and forceful section that gives a synthetic answer to the purpose of the study. The limitations of the study are adequate. A thorough and systematic review of the references section is recommended, trying to comply with the editorial standards of the journal.
We thank the Reviewer for their positive comments on the Discussion.
This review is specifically focused on breathwork interventions for patients clinically diagnosed with anxiety disorders. Other studies and systematic reviews cited throughout the Discussion examine the effects of breathwork interventions in non-clinical populations, including groups at risk (see lines 478-479 and 498-499). Deeper discussion of breathwork interventions that includes these populations goes beyond the scope of our review.
We added a conclusion section, following the reviewer’s recommendation.
We performed a systematic review of the reference sections of each of the included studies (see Methods).